# EPHB4-RASA1-Mediated Negative Regulation of Ras-MAPK Signaling in the Vasculature: Implications for the Treatment of EPHB4- and RASA1-Related Vascular Anomalies in Humans

**DOI:** 10.3390/ph16020165

**Published:** 2023-01-23

**Authors:** Di Chen, Martijn A. Van der Ent, Nathaniel L. Lartey, Philip D. King

**Affiliations:** Department of Microbiology and Immunology, University of Michigan Medical School, Ann Arbor, MI 48109, USA

**Keywords:** Ephrin receptor B4, RASA1, Ras-MAPK pathway, capillary malformation arteriovenous malformation, lymphatic malformation, angiogenesis, vascular valves

## Abstract

Ephrin receptors constitute a large family of receptor tyrosine kinases in mammals that through interaction with cell surface-anchored ephrin ligands regulate multiple different cellular responses in numerous cell types and tissues. In the cardiovascular system, studies performed in vitro and in vivo have pointed to a critical role for Ephrin receptor B4 (EPHB4) as a regulator of blood and lymphatic vascular development and function. However, in this role, EPHB4 appears to act not as a classical growth factor receptor but instead functions to dampen the activation of the Ras-mitogen activated protein signaling (MAPK) pathway induced by other growth factor receptors in endothelial cells (EC). To inhibit the Ras-MAPK pathway, EPHB4 interacts functionally with Ras p21 protein activator 1 (RASA1) also known as p120 Ras GTPase-activating protein. Here, we review the evidence for an inhibitory role for an EPHB4–RASA1 interface in EC. We further discuss the mechanisms by which loss of EPHB4–RASA1 signaling in EC leads to blood and lymphatic vascular abnormalities in mice and the implications of these findings for an understanding of the pathogenesis of vascular anomalies in humans caused by mutations in *EPHB4* and *RASA1* genes. Last, we provide insights into possible means of drug therapy for EPHB4- and RASA1-related vascular anomalies.

## 1. Introduction

Ephrin receptors (EPHR) represent the largest family of receptor tyrosine kinases (RTK) in mammals that regulate such diverse processes as cell migration, proliferation, survival and differentiation in multiple different cell types and tissues [1,2,3]. Of the two classes of EPHR, there are nine EPHA receptors that bind five different Ephrin-A ligands and five EPHB receptors that bind three different Ephrin-B ligands [4,5]. EPHR are typical RTK that comprise an extracellular region with ligand binding and fibronectin homology domains, a short transmembrane spanning domain, and an intracellular region that contains a juxtamembrane (JM) regulatory sequence, a protein tyrosine kinase (PTK) domain and a C-terminal sterile alpha motif (SAM) domain (Figure 1) [1,2,3,4,5]. EPHR can activate multiple different intracellular signaling pathways that couple cell surface ligand recognition events to downstream cellular responses. These include activation of small GTP-binding proteins of the Rho and Ras families and activation of the phosphatidyl inositol 3-kinase (PI3K)-AKT signaling pathway [1,2,6]. The activation or inhibition of these pathways, particularly Ras signaling, can vary depending on the cellular context [1,2,6]. 

In the circulatory system, in vitro analyses, human genetic studies and mouse gene-targeting studies have each pointed to a pivotal function for EPHB4 in particular as a regulator of vascular development and function (see below). Moreover, in this role, there is now significant evidence that EPHB4 communicates with Ras p21 protein activator 1 (RASA1), also known as p120 Ras GTPase-activating protein (p120RasGAP), to inhibit Ras activation in endothelial cells (EC). Herein, we review this evidence and provide insights into possible molecular therapies for vascular disorders arising from impaired EPHB4-RASA1 signaling in humans.

## 2. EPHR-Mediated Inhibition of Ras Signaling In Vitro

RTK activate Ras through recruitment of Ras guanine nucleotide exchange factors (GEF) to cell membranes that switch membrane-tethered Ras from its inactive GDP-bound state to its active GTP-bound state [7,8]. In its GTP-bound state, Ras triggers the activation of downstream signaling pathways including the mitogen-activated protein kinase (MAPK) pathway that culminates in the phosphorylation and activation of extracellular signal-regulated serine/threonine protein kinases (ERKs) [9]. Activated ERKs drive cellular responses through multiple mechanisms including phosphorylation and activation of nuclear transcription factors. 

Several studies performed in cell lines in vitro have shown that EPHR can inhibit activation of the Ras-MAPK pathway triggered by distinct RTK [10,11,12,13,14,15,16,17]. This has been demonstrated for both EPHA and EPHB class receptors, including EPHB4, and in a variety of cell types including myoblasts, neurons, transformed cells and EC. In some studies, the ability of EPHR to inhibit Ras-MAPK activation was shown to be dependent upon expression of functional RASA1 (p120 RasGAP) [10,15,16]. 

RASA1 is one member of family of ten different RasGAPs in mammals [18]. It comprises two N-terminal Src homology-2 (SH2) domains separated by an SH3 domain, followed by pleckstrin homology (PH) and protein kinase C2 homology (C2) domains and a C-terminal catalytically active GAP domain (Figure 1). As a RasGAP, RASA1 inactivates Ras by greatly augmenting its ability to hydrolyze bound GTP to GDP. Expression of a truncated form of RASA1 in neuronal cells blocked an ability of EPHB receptors to inhibit Ras-MAPK activation [10]. In addition, knockdown of RASA1 in myoblasts prevented an ability of EPHA receptors to inhibit Ras-MAPK activation induced by insulin growth factor receptor 1 [15]. For EPHB4, knockdown of RASA1 in the human umbilical vein EC (HUVEC) blocked an ability of EPHB4 to inhibit activation of the Ras-MAPK pathway [16]. Interestingly, although EPHB4 inhibits Ras-MAPK signaling in EC and EC proliferation through a RASA1-depednedent mechanism, in other cell types such as hepatic stellate cells or MC7 breast cancer cells, EPHB4 activates the Ras-MAPK pathway resulting in increased proliferation [16,19]. In MC7 cells, EPHB4 couples to the Ras-MAPK pathway through protein phosphatase 2A (PP2A) because knockdown of PP2A blocks the EPHB4-induced Ras-MAPK response [16]. An ability of EPHB4 to inhibit Ras-MAPK signaling in EC but activate the same pathway in MC7 cells cannot be explained on the grounds of differential expression of RASA1 and PP2A because both are well expressed in the respective cell types. Hence, the basis for this difference remains to be determined. 

Notwithstanding the cell context-dependent signaling activity of EPHB4, the above studies are consistent with the notion of an EPHR–RASA1 inhibitory axis that regulates Ras-MAPK activation in different cell types. However, evidence for this mechanism as a regulator of Ras-MAPK activation in vivo is thus far limited to EPHB4 in EC. 

## 3. Human Vascular Anomalies Caused by *EPHB4* and *RASA1* Mutations

Germline inactivating mutations of *EPHB4* and *RASA1* genes result in the same vascular anomalies in humans. This finding supports the idea that EPHB4 and RASA1 act together in the same pathway performing essentially the same function in the vasculature in vivo.

### 3.1. Capillary Malformation–Arteriovenous Malformation

Capillary malformation–arteriovenous malformation (CM–AVM) is an inherited autosomal dominant vascular anomaly with characteristic cutaneous CM and, in approximately one third of individuals, additional fast flow vascular lesions including AVMs and arteriovenous fistulas that can be life-threatening depending upon location [20,21,22]. In rare cases, lymphatic vessel disorders are also present which include lymphatic malformation and disordered lymphatic vessel flow, chylous ascites and chylothorax (accumulation of lymph fluid in the peritoneal and pleural cavities, respectively) [21,23,24,25]. The first gene reported to be affected in CM–AVM was *RASA1* [26]. The vast majority of *RASA1* mutations in CM–AVM are nonsense mutations, splice site substitutions and frame shift mutations that result in premature stop codons leading to nonsense-mediated RNA decay and predicted total loss of RASA1 protein directed by the mutated germline allele. A minority of mutations are missense mutations that are predicted to affect RASA1 function if not abundance (see below). To explain the autosomal dominant nature of inheritance as well as the sporadic nature of lesions in CM–AVM, a second hit model of CM–AVM pathogenesis was proposed in which inactivating somatic mutation of the inherited wild-type *RASA1* allele in EC during development is conceived to result in RASA1 null EC that drive lesion development [20]. This possibility was first convincingly demonstrated in a study in which a nonsense *RASA1* mutation that was in trans to the inherited *RASA1* mutation was identified in EC derived from a micro-AVM in a patient with CM–AVM but was absent from EC from non-lesional tissue in the same patient [27]. 

*RASA1* gene mutations account for approximately 70 percent of CM–AVM cases. More recently, *EPHB4* gene mutations have been identified to be responsible for the remaining 30 percent of CM–AVM cases [28]. Approximately half of the *EPHB4* mutations in CM–AVM are predicted to result in loss of protein expression consequent to nonsense-mediated RNA decay, whereas the remaining *EPHB4* mutations are missense mutations. CM–AVM cases caused by *RASA1* or *EPHB4* mutations have been named CM–AVM1 and CM–AVM2 respectively to distinguish the two genetic causes. However, CM–AVM1 and CM–AVM2 are phenotypically indistinguishable except for the occurrence of blood vascular telangiectasias in some CM–AVM2 patients. 

### 3.2. Other Vascular Anomalies

The vein of Galen malformation (VGAM) is the most common AVM in pediatric patients [29,30]. Although VGAM has been described in CM–AVM, other cases may present without the CM that are pathognomonic for CM–AVM. Inactivating germline mutations of *EPHB4* and *RASA1* have both been reported in VGAM [21,31,32,33,34]. In addition, inactivating mutations of *EPHB4* and *RASA1* have both been reported in patients with lymphatic vessel disorders without CM–AVM. For *EPHB4*, these include lymphatic-related hydrops fetalis (LRHF, accumulation of fluid in the fetus because of lymphatic dysfunction), central conducting lymphatic anomaly (a lymphatic vessel flow disorder) and lymphedema (tissue edema consequent to lymphatic dysfunction) [35,36,37,38]. Similarly, germline inactivating *RASA1* mutations have been reported in LRHF and lymphedema [39,40]. 

## 4. Mouse Genetic Studies

Mouse genetic studies provide further evidence for an inhibitory EPHB4–RASA1 axis in the vasculature in vivo. With few exceptions, genetic disruption of EPHB4 or RASA1 results in similar if not identical phenotypes. Each of these is described below.

### 4.1. Developmental Angiogenesis

Constitutive EPHB4-deficient and RASA1-deficient mice both die at embryonic day 10.5 (E10.5) of development from cardiovascular defects. The salient phenotype in both models is blocked developmental angiogenesis in which primitive vascular plexuses that are generated through the process of vasculogenesis fail to become remodeled into hierarchical arterial-capillary-venous networks [41,42]. The same phenotype was observed in mice constitutively deficient in the EPHB4 ligand, ephrin B2 [41,43,44]. In addition, the same phenotype was noted in R780Q RASA1 knockin mice that express a form of RASA1 that is unable to promote Ras hydrolysis of GTP, but which would theoretically retain all other putative Ras-independent functions [45]. This latter finding shows that impaired developmental angiogenesis in constitutive RASA1-deficient mice results from an inability of RASA1 to promote Ras inactivation specifically. By extension, vascular anomalies in CM–AVM1 are also likely to be consequent to an inability of RASA1 to inactivate Ras and not to loss of a Ras-independent function of RASA1, e.g., Rho family GTPase regulation as suggested by others [46].

To study the mechanism by which loss of EPHB4 and RASA1 impairs developmental angiogenesis, Chen et al. used conditional EPHB4- or RASA1-deficient mouse models to disrupt the respective genes within EC at E13.5 when vasculogenesis has ceased and developmental angiogenesis predominates [47,48]. In both models, by E18.5, extensive cutaneous hemorrhage was observed that was associated with apoptosis of blood vascular EC (BEC) and loss of lymphatic EC (LEC). In both cases, EC failed to export the vascular basement membrane protein, collagen IV, that instead was trapped within the EC endoplasmic reticulum (ER). The accumulated collagen triggered EC apoptosis though induction of an unresolved unfolded protein response. In addition, the much-reduced amounts of collagen IV in vascular basement membranes resulted in detachment of EC and apoptosis by anoikis. Anoikis (Greek for “without a home”) is the default form of apoptosis that results when a normally adherent cell fails to attach to a substratum [49]. The small molecular weight chaperone, 4 phenyl butyric acid (4PBA), that had been shown to promote folding and export of ER-trapped misfolded point mutant forms collagen IV beforehand, rescued collagen IV export and EC apoptosis in induced EPHB4- and RASA1-deficient embryos [47,48,50,51]. This finding demonstrated that collagen IV retention in the ER was likely a result of collagen IV misfolding (Figure 2).

Consistent with the model of EPHB4 as a negative regulator of Ras-MAPK signaling that acts through communication with RASA1, loss of EPHB4 or RASA1 both resulted in greatly augmented activation of ERK MAPK in EC during developmental angiogenesis [47,48]. Moreover, drugs that inhibit MAPK activation also allowed normal collagen IV export from EC and rescued EC apoptosis showing that dysregulated Ras-MAK activation was the key initiating event in the development of vascular phenotypes in induced EPHB4- and RASA1-deficient embryos [47,48] (Figure 2).

To understand how dysregulated Ras-MAPK activation might lead to collagen IV misfolding, proteomic analyses were performed upon RASA1-deficient and control cutaneous EC during developmental angiogenesis [48]. Intriguingly, of seven members of a family of nine collagen proline and lysine hydroxylases that could be detected, all were increased in abundance in RASA1-deficient compared to control EC. Hydroxylation of collagens on prolines and lysine residues in the ER promotes folding of collagen monomers into trimeric promoters and their export from the cell via the COPII secretion pathway [52,53,54]. However, excessive hydroxylation may impair collagen folding and cellular export. In agreement with a model in which increased abundance of collagen proline and lysine hydroxylases in the EC ER results in over-hydroxylation of collagen IV monomers that blocks collagen IV export, drugs that inhibit all members of the collagen proline and lysine hydroxylase family (part of the larger group of 2 oxoglutarate-dependent oxygenases) also rescued collagen IV export and EC apoptosis during developmental angiogenesis in EPHB4- and RASA1-deficient embryos [48,55] (Figure 2). 

### 4.2. Development and Maintenance of Vascular Valves

The collection of lymphatic vessels and veins that are invested with intraluminal semilunar valves can facilitate directional fluid flow toward the blood vasculature and heart, respectively [56,57]. In addition, specialized valves, known as lymphovenous valves, prevent backflow of venous blood into lymphatic vessels at their point of connection with the venous vasculature in the region of intersection of jugular veins, the subclavian vein and the superior vena cava [57,58,59]. Studies of conditional EPHB4- and RASA1-deficient mice have revealed that EPHB4 and RASA1 are both required for the development all three types of valve [36,60,61,62,63]. Furthermore, all three types of valve fail to develop properly in embryos induced to express R780Q RASA1 alone in the vasculature [60,62]. Consistent with this, blocked development of lymphovenous valves and venous valves in induced RASA1-deficient embryos can be rescued by inhibitors of Ras-MAPK signaling [60]. In addition, drugs that inhibit the activity of collagen IV-modifying proline and lysine hydroxylases, and 4PBA can rescue development of lymphovenous valves and venous valves in induced RASA1-deficient embryos [60]. Therefore, a similar mechanism invoked to explain impaired developmental angiogenesis likely also accounts for impaired development of lymphovenous and venous valves. In this regard, an inability of valve leaflet-forming EC to deposit collagen IV into the extracellular matrix core of the developing valve leaflet would be expected to lead to apoptotic death of leaflet cells and block valve leaflet elongation. 

As well as development, maintenance of lymphatic vessel valves in adult mice also depends upon EPHB4 and RASA1 and catalytically active RASA1 specifically [62] (and unpublished). In induced RASA1-deficient adult mice, it is estimated that popliteal lymphatic vessel valve leaflets lose an average of one LEC per week resulting in progressive valve shortening until a threshold is reached at which point valves are unable to prevent fluid backflow [62]. Consequently, mice exhibit grossly disrupted lymphatic flow patterns and eventually die from chylothorax [64]. Similarly, induced EPHB4-deficient mice show abnormal lymphatic vessel flow and at least some mice succumb to chylothorax (unpublished). 

### 4.3. Neonatal and Pathological Angiogenesis

The disruption of either *Eph4* or *Rasa1* genes in mice or the switch to expression of R780Q RASA1 alone in the neonatal period in mice results in impaired retinal angiogenesis as evidenced by reduced EC coverage and fewer vessel branch points [47,48]. Retinas from both types of mice show increased numbers of “empty sleeves” that consist of thin sheaths of collagen IV without EC. These findings are consistent with a function for EPHB4 and RASA1 in export of collagen IV from EC during active angiogenesis. In this regard, an inability of EC to export collagen IV during retinal angiogenesis would result in loss of EC through detachment and apoptotic death resulting in the “empty sleeve” appearance. 

EPHB4 and RASA1 are also required for pathological angiogenesis necessary for the growth of solid tumors in adult mice [47,48]. In the absence of EPHB4 or RASA1, tumor angiogenesis is impaired and is associated with retention of collagen IV in BEC. Significantly, 4PBA restores collagen IV export, tumor angiogenesis and tumor growth [47,48]. These findings represent an additional example of a requirement for EPHB4 and RASA1 for active angiogenesis and provide further evidence that they function in the same molecular pathway in vivo.

### 4.4. Differences between EPHB4- and RASA1-Deficient Mouse Models

Some differences in phenotype between EPHB4-deficient and RASA1-deficient mouse models are apparent. Notably, induced EC-specific disruption of EPHB4 in adult mice results in rupturing and remodeling of cardiac capillaries and cardiac hypertrophy [17]. In contrast, cardiac hypertrophy has not been observed in induced EC-specific RASA1-deficient mice [64]. In addition, it has been reported that induced EC-specific disruption of EPHB4 in postnatal mice leads to loss of junctional integrity between LEC in collecting lymphatic vessels [61]. In contrast, no such loss of LEC junctional integrity has been observed in induced EC-specific RASA1-deficient mice [62,64]. Potentially, such differences could be explained by loss of EPHB4-mediated activation of distinct signaling pathways, such as the PI3K signaling pathway, in EC [17,65]. Alternatively, or in addition, some differences may relate to a loss of reverse signaling through Ephrin B2 in EPHB4-deficient mice [2,66,67].

## 5. Mechanism of EPHB4–RASA1 Communication in the Vasculature

EPHB4 and RASA1 are known to interact physically during EPHB4 signal transduction [11,47]. The interaction is mediated by RASA1 SH2 domain recognition of a pair of autophosphorylated tyrosine residues in the EPHB4 JM region (Figure 3). Therefore, one obvious model that could account for the functional relationship between EPHB4 and RASA1 in the vasculature is that EPHB4 serves to recruit RASA1 to the plasma membrane whereupon it would be juxtaposed to Ras-GTP that would facilitate Ras inactivation. To test this model, Chen et al. generated a knockin *Ephb4* mutant mouse allele that encodes a form of EPHB4 that is unable to bind RASA1 [47]. Because phosphorylation of the same EPHB4 JM tyrosine residues is necessary to switch EPHB4 from an inactive to an active open conformation, simple mutation of these tyrosines to phenylalanine would not be informative, as such a receptor would be devoid of kinase activity, which would complicate interpretation of results [47,68,69,70]. Therefore, in addition to tyrosine to phenylalanine substitutions, two key proline residues in the JM region were substituted for glycine in the mutant receptor [68,69,70]. These additional mutations circumvented a requirement for phosphorylation of JM tyrosine residues for EPHB4 kinase activity [47]. 

Surprisingly, mice that were homozygous for the RASA1-binding deficient EPHB4 receptor showed normal development of the blood vascular system. In addition, mice exhibited normal retinal angiogenesis and pathological angiogenesis toward solid tumors [47]. Therefore, these results demonstrate that whatever the basis for the functional relationship between EPHB4 and RASA1 in the blood vasculature, it is not require physical interaction between the two molecules. Considering alternative possibilities, it is of note that of the small number of *RASA1* missense mutations that have been reported in CM–AVM, most are concentrated within the PH and C2 domains of the protein that are implicated in membrane lipid binding [20,26,34,71]. An alternative model, therefore, is that RASA1 is targeted to Ras-GTP though recognition of membrane lipids generated during course of EPHB4 signaling (Figure 3). However, experimental evidence for this model awaits.

## 6. Pathogenesis of Vascular Anomalies in Patients with *EPHB4* and *RASA1* Mutations

How second hit mutation of *EPHB4* or *RASA1* genes in patients with inherited mutated *EPHB4* or *RASA1* genes, respectively, gives rise to vascular lesions is uncertain. In most cases, it seems reasonable to propose that the exact nature of lesions, i.e., CM, AVM or lymphatic abnormalities or combinations thereof, depends upon the location of the second hit or hits, i.e., which vascular bed, as well as the time during development that the second hit(s) mutation occurs. However, reconciling how in humans second hit mutation in EC during developmental angiogenesis leads to vascular malformation, whereas in mice loss of EC EPHB4 or RASA1 during developmental angiogenesis results in apoptotic death of EC, is more challenging. One possible explanation for this apparent difference is that in humans, EC that have acquired a second hit mutation are rescued from anoikis on the basis that adjacent EC that have not acquired a second hit mutation are able to export collagen IV normally for deposition in the basement membrane. On the other hand, apoptotic death of EC resulting from an unfolded protein response would not be subverted by an ability of adjacent EC without second hit mutations to continue to engage in collagen IV export. In the case that EC death is substantively rescued through avoidance of anoikis, Ras-MAPK signaling must drive vascular malformation through some undetermined mechanism unrelated to export of collagen IV (Figure 4).

An alternative explanation for the origin of AVMs relates to a potential inability to remodel individual vessels in primitive vascular plexuses. In this regard, should a second hit mutation occur early enough in blood or lymphatic vasculogenesis, it is possible that the majority if not all EC within an individual vessel of a primitive plexus may not express any functional EPHB4 or RASA1. In such a circumstance, remodeling of that vessel by sprouting angiogenesis would not be possible because the demand for de novo synthesis of collagen IV by EC engaging in this event would lead to their apoptotic death (Figure 4). The result would be direct connection between arteries and veins without an intervening capillary bed. For the lymphatic vasculature, it is also possible that, if second hit mutations were to occur early enough, all putative valve-forming cells within a zone of a collecting lymphatic vessel where valve formation would be required would be EPHB4 or RASA1 null. As such, valvulogenesis would not proceed as this would also necessitate collagen IV synthesis. In this way, second hit mutation could contribute to some lymphatic pathologies.

Last, it is of note that in some families with *EPHB4* mutations, affected individuals in different generations present only with lymphatic abnormalities, i.e., LRHF or lymphedema [36,37,38]. This finding is difficult to explain the context of the second hit model of disease expression because it demands that the second hit consistently occurs in lymphatic but not blood vascular beds in each affected individual in these families. Nor is there any evidence that mutant forms of EPHB4 in these families show dominant negative behavior that might otherwise provide an alternative explanation for the lymphatic only phenotype. Hence, the origin of lymphatic abnormalities in these patients remains to be determined.

## 7. Prospects for Drug Therapy of Vascular Anomalies Caused by EPHB4 or RASA1 Mutation

Based on studies performed in mice, drugs that dampen the activation of MAPK or promote the folding of collagen IV in the EC ER, either through blockade of collagen IV proline or lysine hydroxylases or through direct action on collagen IV itself, have potential for the prevention and treatment of vascular anomalies in patients with inherited *EPHB4* and *RASA1* gene mutations. However, the utility of each will depend upon which of the models of vascular pathogenesis outlined above is likely to apply. In the case that EC with second hit mutations can be rescued from anoikis by proximal EC without second mutations, drugs that promote collagen IV folding are not likely to be of benefit. However, in the alternative hypothesis of impaired remodeling of primitive vascular plexuses, these drugs and MAPK pathway inhibitors could be useful. 

Another consideration is the time at which drugs would need to be administered for preventative or therapeutic effect. In most cases, vascular lesions in patients arise during development and are present at birth, although can be exacerbated during postnatal life by events such as puberty or pregnancy. Therefore, prevention of lesion development would require drug administration during fetal development. The extent to which maintenance or growth of lesions requires ongoing dysregulated signal transduction through the Ras-MAPK pathway is unknown. Thus, it is unclear if drug intervention would be effective for the treatment of established vascular lesions. However, this is also likely to depend upon which model of pathogenesis applies. In the case of anoikis rescue, MAPK pathway inhibitors may still have utility. Understanding which model of pathogenesis applies awaits the development of additional mouse models that recapitulate the second hit that is considered to account for most vascular anomalies arising from *EPHB4* or *RASA1* mutation. 

## 8. Summary

A preponderance of evidence from in vitro and in vivo studies now supports the view that the principal function of EPHB4 in the vasculature is to negatively regulate the activation of the Ras-MAPK pathway in EC and that in this role RASA1 is the EPHB4 effector. Loss of EPHB4 or RASA1 function in EC results in dysregulated Ras-MAPK signaling that drives the development of vascular abnormalities in CM–AVM and other vascular disorders. In mice, the dysregulated Ras-MAPK signaling inhibits EC export of collagen IV, which at least in part is responsible for the development of vascular abnormalities. Whether or not the same mechanism is responsible for the development of vascular abnormalities in humans remain to be established. Conceivably, blocked EC export of collagen IV during developmental angiogenesis could prevent remodeling of primitive vascular plexuses into hierarchical arterial-capillary-venous networks, which could account for some types of vascular abnormalities resulting from loss of RASA1 or EPHB4 function in humans. Alternatively, because loss of RASA1 or EPHB4 function is initially expected to be localized to individual EC in human vascular disorders, it is possible that adjacent RASA1- or EPHB4-sufficient EC can rescue RASA1- or EPHB4- null EC from apoptosis through provision of collagen IV in trans. In this latter situation, dysregulated Ras-MAPK signaling may drive vascular abnormalities through yet to be characterized mechanisms. Prospectively, drugs that inhibit the Ras-MAPK pathway or promote EC export of collagen IV could be of benefit for the prevention and treatment of patients with vascular anomalies resulting from inherited and somatic mutations in *EPHB4* or *RASA1* genes. 

## Figures and Tables

**Figure 1 pharmaceuticals-16-00165-f001:**
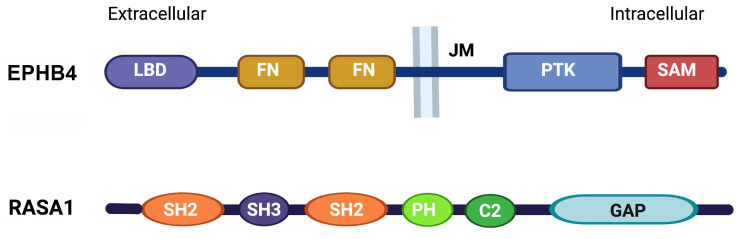
Domain structure of EPHB4 and RASA1. LBD, ligand-binding domain; FN, Fibronectin homology; JM, juxtamembrane region; PTK, protein tyrosine kinase domain; SAM, sterile alpha motif domain; SH2 and SH3, Src homology-2 and 3 domains; PH, pleckstrin homology domain; C2, protein kinase C2 homology domain; GAP, RasGTPase-activating domain.

**Figure 2 pharmaceuticals-16-00165-f002:**
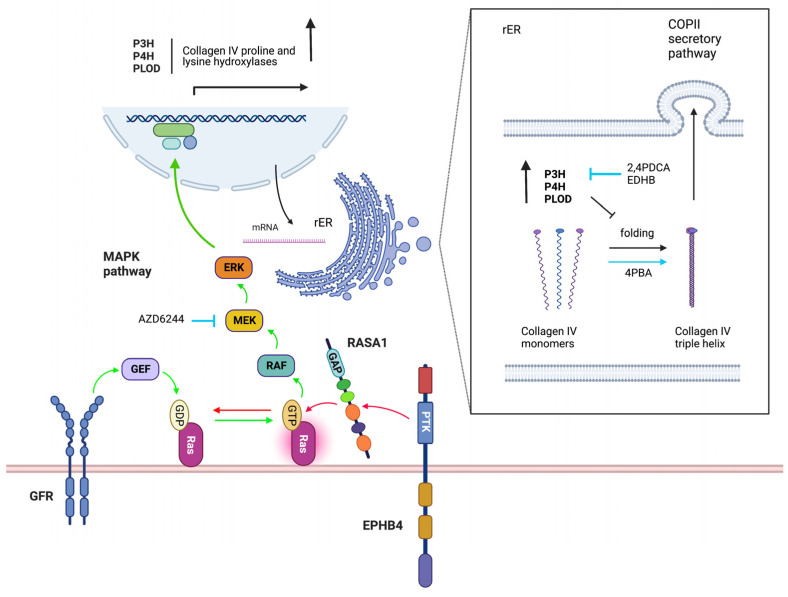
Control of EC collagen IV secretion by EPHB4 and RASA1. During developmental angiogenesis, EPHB4 cooperates with RASA1 to limit GFR-induced Ras activation in EC (red arrows). In the absence of functional EPHB4 and RASA1, dysregulated Ras-MAPK activation results in increased transcription and abundance of collagen P3H and P4H proline and PLOD lysine hydroxylases. In the rough ER (rER), the increased abundance of P3H, P4H and PLODs leads to excessive hydroxylation of collagen IV monomers that inhibits their folding and assembly into triple helical collagen IV protomers that are normally exported from the cell by the COPII secretory pathway. Improperly folded collagen IV is retained within the ER resulting in EC apoptosis. Drugs that inhibit the MAPK pathway (AZD6244), drugs that inhibit collagen IV proline and lysine hydroxylases (2,4PDCA and EDHB) and drugs that promote collagen IV folding in the ER (4PBA) all rescue collagen IV export and EC apoptosis in EPHB4- and RASA1-deficient embryos (blue lines and arrows).

**Figure 3 pharmaceuticals-16-00165-f003:**
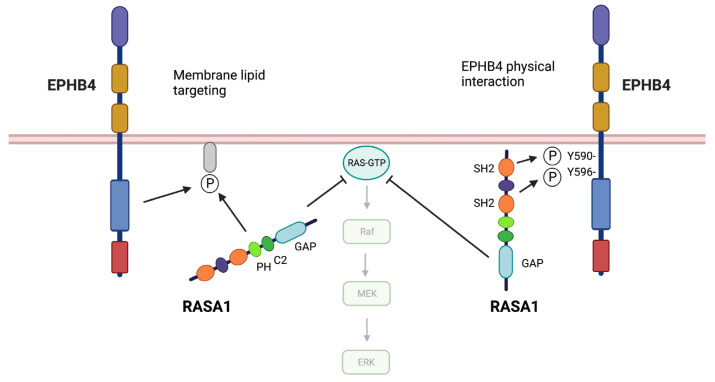
Models of EPHB4 and RASA1 cooperation in EC. Left, EPHB4 signaling generates membrane lipids that are ligands of the RASA1 PH domain leading to RASA1 membrane localization and juxtaposition to Ras-GTP. Right, RASA1 membrane localization is mediated through SH2 domain recognition of phosphorylated tyrosine residues in the EPHB4 JM region. Note that this latter mechanism of EPHB4–RASA1 interplay is not required for developmental angiogenesis or neoangiogenesis in mice.

**Figure 4 pharmaceuticals-16-00165-f004:**
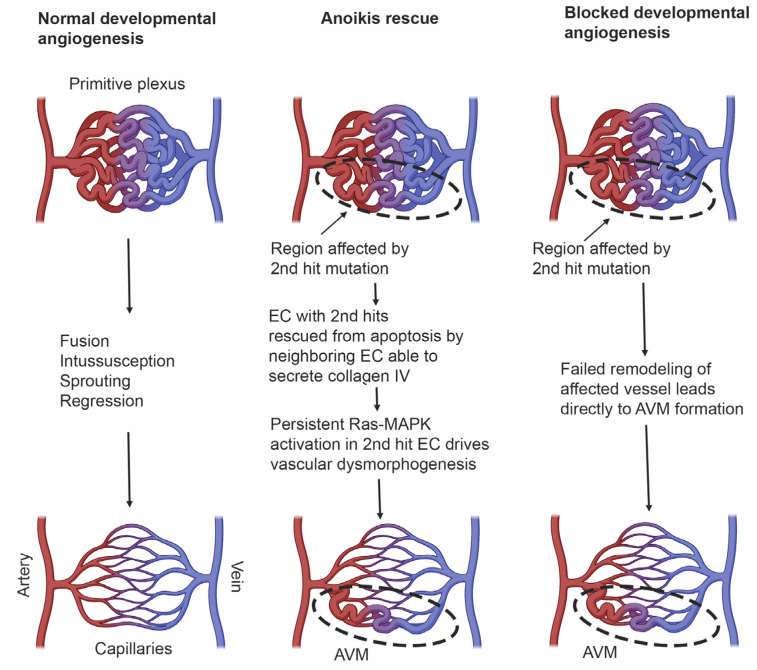
Models of AVM formation resulting from germline plus 2nd hit mutation in *EPHB4* and *RASA1* genes. See text for details.

## Data Availability

Data sharing is not applicable in this paper.

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
