# Peer review of "EPHB4-RASA1-Mediated Negative Regulation of Ras-MAPK Signaling in the Vasculature: Implications for the Treatment of EPHB4- and RASA1-Related Vascular Anomalies in Humans"

_pharmaceuticals, 2023, doi:10.3390/ph16020165_

Round 1

Reviewer 1 Report

Dear authors,

it's a great focus on the subject but i would like to tell you that this review needs modifications before considered, please see the following list of points which need to be considered:

-The abstract is not well organized for reaching conclusions

-Introduction requires more organization

-Inapparent key points in the document make it unique

- the Summary must be more commented.

I congratulate the authors for the good review.

My best regards

Author Response

We thank the reviewer for their comments. Please see responses below. Changes to the manuscript are indicated by yellow highlighting.

The abstract is not well organized for reaching conclusions

Response: We have expanded the abstract

Introduction requires more organization

Response: We wish to have a concise introduction to a focussed topic. In response to another reviewer, we have inserted additional references in the introduction section

Inapparent key points in the document make it unique

Response: In the abstract and summary, we have now covered all main points in the manuscript (in summary form)

The Summary must be more commented.

Response: The summary has now been expanded

Reviewer 2 Report

Manuscript titled "EPHB4-RASA1-mediated negative regulation of Ras-MAPK signaling in the vasculature: implications for the treatment of vascular malformations arising from inherited mutations in the EPHB4-RASA1 pathway." is well written manuscript and has important information on valve pathology and biochemistry. It can be capped without nay changes.

Author Response

(The manuscript) is well written manuscript and has important information on valve pathology and biochemistry. It can be capped without any changes.

Response: We thank the reviewer for their comment. We have made some changes in response to comments from the other reviewers.

Reviewer 3 Report

Your manuscript presents an interesting review, but to the improvement of the quality of the manuscript, I suggest respecting some observations:

1. For all terms such as in vitro or in vivo, please add in italic writing in vitro or in vivo

2. RASA1 functions in the EPHB4 signaling pathway to suppress endothelial mTORC1. Activity vascular malformations are linked to mutations in RAS p21 protein activator 1 (RASA1, also known as p120RasGAP). Due to the global expression of this gene, please develop the idea that EPHB4 can inhibit Ras-MAPK signaling in endothelial cells but not in hepatic stellate or transformed cells, with more references. Which types of transformed cells do you mean? Why do you think that mechanism is limited only to endothelial cells?

3. May EphB4 promote or suppress the Ras/MEK/ERK pathway in a context-dependent manner in the function of different cells? Please add more references about this.

4. If targeting EphB4 treatment in vascular malformations and patients also presents various types of malign affections may be conflicting effects on the cancer cell and endothelial cell compartments. If yes, please add some observations and references about this.

Author Response

We thank the reviewer for their comments. Please see responses below. Changes to the manuscript are indicated by yellow highlighting.

For all terms such as in vitro or in vivo, please add in italic writing in vitro or in vivo

Response: completed.

RASA1 functions in the EPHB4 signaling pathway to suppress endothelial mTORC1. Activity vascular malformations are linked to mutations in RAS p21 protein activator 1 (RASA1, also known as p120RasGAP). Due to the global expression of this gene, please develop the idea that EPHB4 can inhibit Ras-MAPK signaling in endothelial cells but not in hepatic stellate or transformed cells, with more references. Which types of transformed cells do you mean? Why do you think that mechanism is limited only to endothelial cells?

Response: We have now addressed these questions and have inserted extra text in the relevant part of section 2 (p.2-3.)

May EphB4 promote or suppress the Ras/MEK/ERK pathway in a context-dependent manner in the function of different cells? Please add more references about this.

Response: Yes there are different functional outcomes depending upon cell type. This information has now been added to section 2 (p2-3).

If targeting EphB4 treatment in vascular malformations and patients also presents various types of malign affections may be conflicting effects on the cancer cell and endothelial cell compartments. If yes, please add some observations and references about this.

Response: We thank the reviewer for this comment but this really is not a problem since vascular anomalies of this sort almost always arise during embryonic development. Therefore, in the vast majority of cases there should not be a conflict with regards different effects of drugs upon tumors versus vascular development. Since there is no direct data on this we have elected not to comment.

Reviewer 4 Report

Di chen et al. reviewed a large amount of literature and systematically sorted out the EPHB4-RASA1-mediated negative regulation of Ras-MAPK 2 signaling in the vasculature. The review is well-written and deepens the understanding of the means of drug therapy for patients with inherited vascular anomalies caused by mutations in 20 EPHB4 and RASA1 genes. This work is very meaningful, but there are still some key points that need attention.

1. As the EPHB4-RASA1 pathway is an emerging pathway, it is necessary to explain how it affects inherited vascular anomalies in the conclusion section also.

2. The readability needs to be improved, especially the Prospects for drug therapy of vascular anomalies caused by EPHB4 or RASA1 mutation.

3. This review reviewed a large amount of literature but lacked the author's point of view and emphasis.

4. The conclusion lacks depth and hence needs to be reframed to exhibit the significance of current research in a more scientific manner.

5. The title is not clear. The title should be changed and should be precise.

6. In the introduction there are relatively few references cited. More references would improve the quality of the review.

Author Response

We thank the reviewer for their comments. Please see responses below. Changes to the manuscript are indicated by yellow highlighting.

As the EPHB4-RASA1 pathway is an emerging pathway, it is necessary to explain how it affects inherited vascular anomalies in the conclusion section also.

Response: We have expanded the conclusion section accordingly

The readability needs to be improved, especially the Prospects for drug therapy of vascular anomalies caused by EPHB4 or RASA1 mutation.

Response: We respectfully submit that the readability is exactly as intended, including the drug therapy section, which is, by nature of the topic, speculative at this point. The manuscript has been written mostly by the senior author whose first language is English.

This review reviewed a large amount of literature but lacked the author's point of view and emphasis.

Response: Our intent is to be as factual as possible and not to express opinion unnecessarily. An exception is section 6, where we present different models of vascular malformation pathogenesis that may be viewed as the opinion of the authors. Further commitment to one model or another is not possible based upon existing evidence.

The conclusion lacks depth and hence needs to be reframed to exhibit the significance of current research in a more scientific manner.

Response: The conclusion has been expanded.

The title is not clear. The title should be changed and should be precise.

Response: We have abbreviated the title somewhat and consider it to be clear and as intended.

In the introduction there are relatively few references cited. More references would improve the quality of the review.

Response: We have added more references to the introduction.

Round 2

Reviewer 4 Report

Authors have incorporated all the suggested changes that has been suggested to them hence the revised manuscript can now be accepted in its current form